# Choline Chloride/Urea Deep Eutectic Solvents: A Promising Reaction Medium for the Synthesis of Bio-Based Poly(hydroxyurethane)s

**DOI:** 10.3390/molecules27134131

**Published:** 2022-06-27

**Authors:** Guanfei Shen, Bruno Andrioletti

**Affiliations:** ICBMS-UMR CNRS 5246, Campus LyonTech la Doua, Université Claude Bernard-Lyon 1, Université de Lyon, 43 Boulevard du 11 Novembre 1918, CEDEX, 69622 Villeurbanne, France; guanfei.shen@univ-lyon1.fr

**Keywords:** non-isocyanate polyurethanes, bio-based, aminolysis, deep eutectic solvents

## Abstract

The development of more sustainable and eco-friendly polymers has attracted much attention from researchers over the past decades. Among the different strategies that can be implemented towards this goal, the substitution of the toxic reagents/monomers often used in polyurethane chemistry has stimulated much innovation leading to the development of the hydroxylated version of PURs, namely, the poly(hydroxyurethane)s (PHURs). However, some PHURs remain far from being sustainable as their synthesis may involve monomers and/or solvents displaying poor environmental impacts. Herein, we report on the use of more sustainable conditions to synthesize the biobased polycarbonates involved in the aminolysis reaction. In addition, we demonstrate that the use of renewable deep eutectic solvents (DESs) can act both as excellent solvents and organocatalysts to promote the aminolysis reaction.

## 1. Introduction

Since their discovery by Otto Bayer in 1947 [1], polyurethanes (PURs) have become the sixth most widely used polymer [2]. Indeed, PURs are especially versatile in various applications such as rigid and flexible foams, clothing, coatings, adhesives, paintings, packaging, elastomers, sealants and so on [3,4].

However, the synthesis of PURs involves the use of isocyanates that are more and more questioned because of health and environmental concerns worldwide [5,6]. Recently, new and safer PURs, called non-isocyanate polyurethanes (NIPURs), have appeared as a promising alternative to replace conventional PURs. Three main routes have been reported for the synthesis of NIPURs: (1) AB-type azide condensation, (2) transurethane reaction, and (3) aminolysis [7,8]. An overview of the different routes usable for the synthesis of NIPUs draws attention to the green synthesis of cyclic carbonate (CC) compounds and the aminolysis reaction [9,10]. Presently, the main focus concerns the reaction between bio-based carbonates and amines, offering an interesting pathway to renewable poly(hydroxyurethanes) (PHURs), an interesting class of NIPURs thanks to the presence of the pending hydroxyl groups resulting from the aminolysis of the CC [11].

Although five cyclic carbonates (5CCs) are less reactive than the six cyclic carbonate analogues, they are easier to synthesize from bio-based compounds using green methodologies. Accordingly, they appear as the best option for the preparation of PHURs [9,12,13]. Thanks to the presence of three reactive hydroxyl groups, glycerol is an interesting C3-bio-based building block to produce highly-added value molecules such as glycerol carbonate (GC), a monofunctional carbonate [14,15]. The presence of the two different functions (hydroxyl and CC) confers GC interesting chemical reactivities and some special physical properties such as high boiling point, high flash point, low flammability, and low toxicity. Those properties are particularly pertinent for industrial applications [9,16].

2,5-Furandicarboxylic acid (FDCA) is a high-value bio-based platform molecule derived from renewable sources that have been acknowledged as a highly valuable feedstock [17,18]. Hence, the elaboration of PHURs incorporating FDCA and GC is very appealing. However, the synthesis of PHURs from the 5CCs reaction generally requires high temperatures and long reaction times that hamper the “green” character of the transformation. Last, but not least, the use of a catalyst is generally necessary to allow the use of a-sterically hindered amines (e.g., α-methyl primary amines) to reach decent molar mass [19,20].

In this work, we report on the reaction of an FDCA-based bis-cyclic carbonate (FDCA-Bis-CC) with a series of variously substituted amines in the choline chloride/urea (ChCl/urea) deep eutectic solvent (DES) that acts both as a green solvent and a catalyst in the aminolysis reaction. To the best of our knowledge, this promising additive-free synthesis of PHUs is unprecedented in the literature (Figure 1).

## 2. Results and Discussion

Generally, the traditional methods used to synthesize the FDCA-bis-CC involve the use of over-stoichiometric quantities of toxic chemicals such as *m*-chloroperoxybenzoic acid or thionyl chloride [21,22,23]. Conversely, inorganic or organic Brönstedt bases or Lewis acids such as distannoxanes have also been reported as efficient catalysts for the transesterification reaction [24,25].

Within the frame of our study aiming at developing the greenest approach, we first explored the use of heterogenous catalysts such as CaO for the transesterification reaction. Toluene was chosen as the solvent and the reaction was carried out at 90 °C. Interestingly, after the reaction, the heterogeneous catalyst could be easily recovered and reused affording a cost-effective green process. Unfortunately, only traces of the expected target product FDCA-Bis-CC (Table 1, entry 1) was isolated. Similarly, other inorganic, alkali bases such as NaOCH_3_ or KOH did not afford the expected product in substantial amounts. NaOCH_3_ (Table 1, entry 2) afforded only traces of the trans-esterified product and KOH (Table 1, entry 3) induced the degradation of the starting material. Fortunately, in the presence of DMAP as an organocatalyst, we could obtain the FDCA-Bis-CC in a 52% isolated yield at 90 °C in toluene after a simple filtration. (Table 1, entry 5).

Thus, we adopted the following synthetic strategy using the DMAP as a catalyst to synthesize the FDCA-bis-CC (Figure 2).

The transesterification being an equilibrated reaction, a slight excess of GC (2.2 equiv.) was used first to ensure the best conversion of DM-FDCA in toluene. Next, the effect of the temperature on the outcome of the transesterification reaction was investigated. In the presence of 10 mol% DMAP, at 80 °C, the expected FDCA-bis-CC was isolated as a white powder in 45% yield by simple precipitation as FDCA-bis-CC is poorly soluble in toluene at room temperature (Table 2, entry 1). Increasing the temperature to 90 °C (Table 2, entry 2), the expected product was isolated in 52% yield by simple filtration. Increasing further the temperature to 100 °C (Table 2, entry 3) allowed us to isolate the FDCA-Bis-CC in a 65% yield. A higher yield was expected at this temperature, but as DM-FDCA is very easily sublimated, part of the starting material was lost in these conditions and induced reproducibility issues. Hence, the optimal compromise between yield and temperature appeared to be 90 °C. In order to improve further the sustainability of the reaction, we investigated the effect of the catalytic charge. In the presence of 5 mol% DMAP at 90 °C, we were happy to observe that the yield of the reaction was not affected (50% yield vs. 52% yield). Unfortunately, decreasing further the catalytic load to 2.5 mol% was detrimental as the expected FDCA-Bis-CC was isolated in only 17% yield (Table 2, entry 5). Another parameter was investigated next: the amount of GC. Interestingly, a slight increase in the excess of GC from 2.2 to 2.5 equiv. allowed isolating the expected decarbonate in 67% yield (Table 2, entry 6), but a plateau was reached as the isolated yield remained pretty much steady when a larger amount of GC was used (Table 2, entry 7). Accordingly, we concluded that the best conditions were 90 °C, 2.5 equiv. GC and 5 mol% catalyst. Based on these optimal conditions, we investigated the effect of the reaction time on the outcome of the reaction. After 48 h, we were pleased to see that we could further increase the efficiency of the reaction and isolate the FDCA-Bis-CC expected product in 58% yield (Table 2, entry 8). Noticeably, when the reaction was carried in a 10 g scale (55 mmol DM-FDCA), an excellent yield of 95% of FDCA-Bis-CC was obtained after a simple filtration and washing with acetone.

Having developed an efficient strategy to prepare the FDCA-Bis-CC on a large scale in the absence of lengthy column chromatographies and the use of highly toxic reagents, we went further and investigated the aminolysis reaction of FDCA-Bis-CC to afford the corresponding PHURs. It is well acknowledged that many parameters can affect the aminolysis reaction and the recent efforts dedicated to optimizing the PHUR synthesis were proven fruitful. Of note, Cornille et al. reported that the nature of the substituents located neighboring the 5CC has a strong influence on its reactivity towards the EDR-148 amine and the following reactivity effect was proposed: ethyl-ester > acetate > trimethylhexanoate > benzoate ≈ ethyl ether > phenyl-ether > butane [26]. Diakoumakos et al. also ranked the reactivity of different amines towards ether-5 CC on the basis of their nucleophilicity and their size. In particular, their study revealed that α-sterically hindered amines display the weaker activity [27]. In order to increase the reactivity of the least nucleophilic amines, various types of catalysts were screened for promoting the aminolysis reaction. Three different mechanisms were proposed. It is possible to (a) increase the electrophilicity of the CC, (b) enhance the nucleophilicity of the amine, or (c) direct the attack on the CC that further reacts as a leaving group [20,28]. In recent years, organocatalysts have demonstrated their prominence in the catalytic aminolysis reaction, among which the TBD and thioureas exhibit the higher effective activity [29,30]. Parallelly, a series of studies on the role of the solvent used during the aminolysis reaction indicated that generally, protic solvents have a positive effect on the reactivity and conversion of CC by increasing the carbonyl positive charge and also by interrupting the intramolecular hydrogen bond network present in the PHUR chains [26,31,32]. Unfortunately, many traditional organic solvents used in chemical reactions are hazardous to the environment and/or present acute and chronic toxicity to humans and a low biodegradability [33]. Conversely, the DESs have been proposed as green, alternative solvents over the past two decades. In particular, natural deep eutectic solvents (NaDESs) are especially appealing as they are easily synthesized from biobased or natural compounds. Hence, most NaDESs are eco-friendly, non-toxic and inexpensive [34,35]. DESs present some remarkable properties such as an excellent dissolution power even for metal oxides and they can act as catalysts and find application in material chemistry [36,37]. Thus, recently, Xue introduced a novel ternary DES able to dissolve the very insoluble lignin allowing the synthesis of a lignin-containing rigid polyurethane foam [38]. In parallel, Han designed a DES based on ChCl and urea supported by molecular sieves as a catalyst for the synthesis of CC from CO_2_ and epoxides [39]. Based on these preliminary results, we investigated the aminolysis of the FDCA-bis-CC using the Jeffamine D-2000 as a nucleophile in a series of green solvents. DMF used as a reference solvent was used as a benchmark in the aminolysis reaction (Table 3).

Using classical conditions (equimolar amounts of diamine and bis-carbonate at 110 °C in DMF), the aminolysis reaction did not proceed as expected as a myriad of products were formed during the reaction (Table 3, entry 1). Switching DMF to glycerol (a protic, biobased solvent; Table 3 entry 2) was not effective either, as the starting material remained untransformed even after 24 h at 110 °C. The initial conditions being unsuccessful, we envisioned the use of DES. Among the almost infinite possibilities of associating a hydrogen bond donor and a hydrogen bond acceptor, we concentrated our efforts on DESs based on ChCl that present the advantage of being a good candidate for assisting the aminolysis reaction and on urea that could activate the carbonyl group of the carbonate function. Other combinations (ChCl/glycerol, ChCl/AcNH_2_, …) were used for assessing the role of the different partners. With the exception of the AcNH_2_/urea DES, which proved to slowly decompose at 110 °C, only the ChCl/urea DES was effective for promoting the aminolysis reaction (Table 3, entry 3). Interestingly, lowering the temperature to 100 °C prevented the reaction to take place (Table 3, entry 4). When other hydrogen bond donors were used at the expense of urea, the reaction did not proceed either (Table 3, entries 5 and 6). Hence, the best conditions for promoting the aminolysis reactions involved the use of ChCl/urea 1: 2 as the solvent at 110 °C.

Accordingly, a series of variously substituted amines were reacted with FDCA-bis-CC in the above-mentioned conditions (Figure 3).

As expected, if the reaction with rather nucleophilic amines afforded the desired PHURs with relatively high molar mass (Table 4, entries 1 and 3) in mild conditions, we were pleased to note that even at a lower temperature (65 °C instead of 110 °C), the reactivity was still preserved opening promises for the use of short polyamines displaying low boiling points and high nucleophilicity. Even less nucleophilic amines such as IPDA or the α-methylated amines were reactive enough to afford the corresponding PHURs (Table 4, entries 4–7), whilst generally with lower *Mn*.

## 3. Conclusions

In this work, we disclose a novel, green, and additives-free method to synthesize PHURs even with α-methyl primary amine that generally displays poor reactivity. This is made possible by the use of the ChCl/urea DES, which is used both as a green solvent and a catalyst. In addition, the high solvent power of the DES allows for the synthesis of polymers displaying a rather high *Mn*. Furthermore, as the bis-carbonate used as the key reagent contains an FDCA moiety that is known to favor biodegradability [40,41,42,43], it is expected that this approach will allow the development of bio-degradable PHURs. Work is currently underway in our Laboratory to expand further the versatility of the approach.

## 4. Experimental Section

### 4.1. Material

FDCA, dimethyl carbonate (DMC), 4-dimethylaminopyridine (DMAP), acetamide (AcNH_2_), ChCl, 3-(aminomethyl)-3,5,5-trimethylcyclohexanamine (IPDA), N,N-Dimethylformamide (DMF), and hexane-1,6-diamine (DA6) were purchased from Aldrich. ethylene glycol bis(2-aminoethyl) ether (EDR-148) was purchased from Merck KGaA company (Darmstadt, Germany). Jeff D-2000, Jeff T-403, and Jeff T-3000 were purchased from Huntsman company (Salt Lake City, UT, USA). All the chemicals were used without further purification. Number-average molecular weight (*Mn*); dispersity (D).

### 4.2. Synthesis of GC

Glycerol (0.4 mol, 36.84 g), DMC (1.4 mol, 126 g), and CaO (6% mmol, 1.35 g) were loaded into a 250 mL round-bottom flask equipped with a condenser and reacted at 75 °C for 5 h. After cooling at room temperature, the catalyst was removed by filtration. MeOH and excess DMC were removed at 45 °C under reduced pressure. The reaction mixture was purified by silica gel chromatography using AcOEt/cyclohexane (2:1) as the eluent. The expected GC was obtained in 86% yield (40.6 g) as a colorless oil [44,45].

^1^H NMR (300 MHz, DMSO) δ 5.28 (t, *J* = 5.6 Hz, 1H), 4.80 (ddd, *J* = 5.9, 4.3, 2.8 Hz, 1H), 4.51 (t, *J* = 8.3 Hz, 1H), 4.30 (dd, *J* = 8.2, 5.8 Hz, 1H), 3.69 (ddd, *J* = 12.6, 5.5, 2.8 Hz, 1H), 3.52 (ddd, *J* = 12.6, 5.7, 3.3 Hz, 1H).

^13^C NMR (75 MHz, DMSO) δ 155.33 (s), 77.15 (s), 65.98 (s), 60.70 (s).

IR (*ν*_max_ cm^−1^) 3411(-OH), 1760 (CC C=O).

MS: [M + H]^+^ 119.1, [M + Na]^+^ 141.0.

### 4.3. Synthesis of Dimethyl Furan-2,5-dicarboxylate (DM-FDCA)

To a solution of 2,5-furandicarboxylic acid (70 mmol, 10.9 g) in dry MeOH (500 mL), 98% sulfuric acid (0.7 mL) was added. The resulting solution was heated at 65 °C for 4 d. The solution was cooled and concentrated in vacuo. The residue was filtered and washed with a saturated aqueous NaHCO_3_ solution (4 × 60 mL) and water until neutral pH. Dimethyl furan-2,5-dicarboxylate (DM-FDCA) was obtained as a white solid in 97% yield (12.5 g) [46,47].

^1.^ H NMR (300 MHz, CDCl_3_) δ 7.19 (s, 2H), 3.89 (s, 6H).

^13^C NMR (75 MHz, CDCl_3_) δ 158.46 (s), 146.69 (s), 118.56 (s), 52.47 (s).

IR (*ν*_max_ cm^−1^): 1720 (ester C=O).

MS [M + H]^+^ 185.0, [M + Na]^+^ 207.0.

### 4.4. Synthesis of Furan-2,5-dicarboxylate Bis Cyclic Carbonate (FDCA-Bis-CC)

Dimethyl Furan-2,5-dicarboxylate (55 mmol, 10.1 g), GC (2.5 equiv., 16.2 g) and DMAP (2 mol%, 134 mg) were dissolved in 200 mL dry toluene. The reaction mixture was stirred at 90 °C for 5 d under a gentle flux of air to remove MeOH. After the reaction, the residue was filtered off and washed with acetone (20 mL × 3) three times. Finally, we obtained the Furan-2,5-dicarboxylate bis cyclic carbonate as a white solid in a 95% yield (18.6 g) [21].

^1^H NMR (400 MHz, DMSO) δ 7.43 (s, 2H), 5.26–5.06 (m, 2H), 4.72–4.46 (m, 6H), 4.40 (dd, *J* = 8.4, 6.1 Hz, 2H).

^13^C NMR (101 MHz, DMSO) δ 156.79 (s), 154.65 (s), 145.81 (s), 119.64 (s), 74.15 (s), 66.05 (s), 64.56 (s).

IR (*ν*_max_ cm^−1^): 1777 (CC C=O), 1735 (ester C=O).

MS: [M + H]^+^ 357.0, [M + Na]^+^ 379.0.

### 4.5. Synthesis of the DESs

The hydrogen bond donor (HBD) and hydrogen bond acceptor (HBA) were added to the round flask in the appropriate proportion. The mixture was stirred and heated slowly until a homogeneous colorless liquid was formed with no visible precipitate.

### 4.6. ChCl/Urea

Using the general procedure, 5 mmol ChCl and 10 mmol urea were stirred at 75 °C until a homogenous solution was obtained [48].

### 4.7. PHURs Syntheses

*General procedure*: FDCA-Bis-CC (1 equiv.) and the appropriate diamine were introduced in a round bottom flask under argon. ChCl/urea (1:2) was added and the reaction mixture was stirred under argon until the monomer was consumed. The reaction mixture was purified by dialysis to remove the DES and the short oligomers.

### 4.8. PHURs Synthesized from FDCA-Bis-CC and EDR-148 at 110 °C in ChCl/Urea

FDCA-Bis-CC (712 mg, 2 mmol), EDR-148 (296 mg, 2 mmol), and 2 mL of DES (ChCl/urea = 1:2) were introduced in a 25 mL flask. The flask was evacuated and re-filled with argon for several times, and finally heated at 110 °C for 24 h. After the reaction, the mixture was poured into a dialysis bag (MWCO = 3500), and extracted three times every three hours with 500 mL water.

^1^H NMR (400 MHz, DMSO) 8.6 (urethane N-H), 7.1 (furanic β H).

^13^C NMR (101 MHz, DMSO) 157 (ester), 155 (urethane), 148 (furanic α C), 114 (furanic β C).

IR (*ν*_max_ cm^−1^): 1723 (ester C=O), 1680 (urethane C=ONO), 1525 (urethane N-H).

*Mn* = 19,849, D = 1.05.

### 4.9. PHURs Synthesized from FDCA-Bis-CC and DA6 in CHCl/Urea

FDCA-Bis-CC (712 mg, 2 mmol), DA6 (232 mg, 2 mmol) and 2 mL of DES (ChCl/urea = 1:2) were introduced in a 25 mL flask. The flask was evacuated and re-filled with argon several times, and finally heated at 110 °C for 40 h. After the reaction was completed, water was used to remove the DES.

^1^H NMR (400 MHz, DMSO) 8.7 (urethane N-H), 7.1 (furanic β H).

^13^C NMR (101 MHz, DMSO) 157 (ester), 155 (urethane), 148 (furanic α C), 114 (furanic β C).

IR (*ν*_max_ cm^−1^): 1785 (terminal carbonate), 1713 (ester C=O), 1650 (urethane C=ONO), 1535 (urethane N-H). 

*M**n* = 18,137, D = 1.08.

### 4.10. PHURs Synthesized from FDCA-Bis-CC and IPDA at 110 °C in ChCl/Urea

FDCA-Bis-CC (712 mg, 2 mmol), IPDA (340 mg, 2 mmol), and 2 mL of DES (ChCl/urea = 1:2) were introduced in a 25 mL flask. The flask was evacuated and re-filled with argon several times, and finally heated at 110 °C for 24 h. After the reaction was completed, water was used to remove the DES.

^1^H NMR (400 MHz, DMSO) 8.4–8.7 (urethane N-H), 7.1 (furanic β H).

^13^C NMR (101 MHz, DMSO) 157 (ester), 155 (urethane), 148 (furanic α C), 114 (furanic β C).

IR (*ν*_max_ cm^−1^): 1810 (terminal carbonate) 1712 (ester C=O), 1651 (urethane C=ONO), 1540 (urethane N-H).

*M**n* = 27,374, D = 1.08.

### 4.11. PHURs Synthesized from FDCA-Bis-CC and Jeffamine D-2000 in CHCl/Urea

FDCA-Bis-CC (712 mg, 2 mmol), Jeffamine D-2000 (4 g, 2 mmol), and 0.5 mL of DES (ChCl/urea = 1:2) were introduced in a 25 mL flask. The flask was evacuated and re-filled with argon several times, and finally heated at 110 °C for 40 h. After the reaction was completed, water was used to remove the DES.

^1^H NMR (400 MHz, DMSO) 8.3–8.6 (urethane N-H), 7.1 (furanic β H).

^13^C NMR (101 MHz, DMSO) 158 (ester), 155 (urethane), 148 (furanic α C), 114 (furanic β C).

IR (*ν*_max_ cm^−1^): 1810 (terminal carbonate), 1720 (ester C=O), 1661 (urethane C=ONO), 1539 (urethane N-H). 

*M**n* = 6784, D = 1.01.

### 4.12. PHURs Synthesized from FDCA-Bis-CC and Jeffamine T-403 in CHCl/Urea

FDCA-Bis-CC (534 mg, 1.5 mmol), Jeffamine T-403 (440 mg, 1 mmol), and 0.5 mL of DES (ChCl/urea = 1:2) were introduced into a glass reactor and heated for 40 h at 110 °C under an Ar atmosphere. After the reaction, water was added to remove the DES.

^1^H NMR (400 MHz, DMSO) 8.3–8.5 (urethane N-H), 7.1–7.4 (furanic β H).

^13^C NMR (101 MHz, DMSO) 158 (ester), 155 (urethane), 148 (furanic α C), 114 (furanic β C).

IR (*ν*_max_ cm^−1^): 1785 (terminal carbonate), 1708 (ester C=O), 1652 (urethane C=ONO), 1539 (urethane N-H). 

*M**n* = 32,113, D = 1.04.

### 4.13. PHURs Synthesized from FDCA-Bis-CC and Jeffamine T-3000 in CHCl/Urea

FDCA-Bis-CC (534 mg, 1.5 mmol), Jeffamine T-3000 (3 g, 1 mmol), and 0.5 mL of DES (ChCl/urea = 1:2) were introduced into a glass reactor and heated for 40 h at 110 °C under an Ar atmosphere. After the reaction, water was added to remove the DES.

^1^H NMR (400 MHz, DMSO) 8.1–8.5 (urethane N-H), 7.1–7.3 (furanic β H)

^13^C NMR (101 MHz, DMSO) 157 (ester), 155 (urethane), 148 (furanic α C), 115 (furanic β C).

IR (*ν*_max_ cm^−1^): 1813 (terminal carbonate), 1720 (ester C=O), 1650 (urethane C=ONO), 1534 (urethane N-H).

*M**n* = 7486, D = 1.01.

## Data Availability

Not applicable.

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
