# Peer review of "Choline Chloride/Urea Deep Eutectic Solvents: A Promising Reaction Medium for the Synthesis of Bio-Based Poly(hydroxyurethane)s"

_molecules, 2022, doi:10.3390/molecules27134131_

Round 1

Reviewer 1 Report

The submitted paper deals with the synthesis of bio-based poly(hydroxyurethane)s from cyclic carbonate, being the derivative of 2,5-furandicarboxylic acid and glycerol, and five different diamines, including α-methyl primary diamines. The synthesis was performed in a green solvent, i.e. the choline/urea deep eutectic solvent. The preparation of the cyclic carbonate was also made without using of toxic chemicals, in the presence of 4-dimethylaminopyridine as a catalyst. Furthermore, the use of this carbonate should favour the biodegradation of the materials obtained.

The paper, presenting a novel, eco-friendly and additives-free method to obtain poly(hydroxyurethane)s, is worth publishing in Molecules after minor corrections. My reservations concern editorial side and specialist terminology.

 1.        According to the present IUPAC recommendations (Pure Appl. Chem. 2014; 86(6):1003-1015) the abbreviation accepted for polyurethane is PUR, not PU.

2.        Abstract:

     The abbreviation “PUs” should be introduced and changed to “PURs”.

PolyHydroxyUrethanes” should be replaced with “poly(hydroxyurethane)s”.

     “poly(carbonates)” should be replaced with “polycarbonates”.

  Deep Eutectic Solvents (DESs)” should be replaced with “deep eutectic solvents”.

3.        Introduction:

     PolyUrethanes” should be changed to “polyurethanes”.

     Deep Eutectic Solvent (DES)” should be replaced with “deep eutectic solvent (DES)”.

4.        Results and discussion

        The names of some compounds are not in keeping with the newest IUPAC recommendations, e. g. 1,6-Hexanediamine (the correct name is hexane-1,6-diamine).

        It is incomprehensible to use capital letters in the names of chemicals, polymers, etc.

        The abbreviations should only be introduced once (where they appear first time) and then applied consistently.

        The abbreviation “m-CPBA” was introduced unnecessarily.

        Some abbreviations (e.g. DMF) were not explained at all.

        “molecular weight” should be replaced with “molar mass”

5.        Material

“Glycerol carbonate (GyC).”?

Author Response

Referee # 1:

The submitted paper deals with the synthesis of bio-based poly(hydroxyurethane)s from cyclic carbonate, being the derivative of 2,5-furandicarboxylic acid and glycerol, and five different diamines, including α-methyl primary diamines. The synthesis was performed in a green solvent, i.e. the choline/urea deep eutectic solvent. The preparation of the cyclic carbonate was also made without using of toxic chemicals, in the presence of 4-dimethylaminopyridine as a catalyst. Furthermore, the use of this carbonate should favour the biodegradation of the materials obtained.

The paper, presenting a novel, eco-friendly and additives-free method to obtain poly(hydroxyurethane)s, is worth publishing in Molecules after minor corrections. My reservations concern editorial side and specialist terminology.

  1. According to the present IUPAC recommendations (Pure Appl. Chem. 2014; 86(6):1003-1015) the abbreviation accepted for polyurethane is PUR, not PU.

Answer: Thank you for your kind suggestion. The changes have been made accordingly in the text.

2.Abstract:

  • The abbreviation “PUs” should be introduced and changed to “PURs”.
  • PolyHydroxyUrethanes” should be replaced with “poly(hydroxyurethane)s”.
  • “poly(carbonates)” should be replaced with “polycarbonates”.
  • “Deep Eutectic Solvents (DESs)” should be replaced with “deep eutectic solvents”.

Answer: Thank you. The changes have been made accordingly in the text.

3.Introduction:

  • “PolyUrethanes” should be changed to “polyurethanes”.
  • “Deep Eutectic Solvent (DES)” should be replaced with “deep eutectic solvent (DES)”.

Answer: Thank you for your kind suggestion. The changes have been made accordingly in the text.

4.Results and discussion

  • The names of some compounds are not in keeping with the newest IUPAC recommendations, e. g. 1,6-Hexanediamine (the correct name is hexane-1,6-diamine).
  • It is incomprehensible to use capital letters in the names of chemicals, polymers, etc.
  • the abbreviations should only be introduced once (where they appear first time) and then applied consistently.
  • The abbreviation “m-CPBA” was introduced unnecessarily.
  • Some abbreviations (e.g. DMF) were not explained at all.
  • “molecular weight” should be replaced with “molar mass”

Answer: Thank you for the comment, we have corrected as suggested. DMF (Dimethyl formamide) is a common organic solvent which abbreviation is admitted even in text books.

  1. Material

“Glycerol carbonate (GyC).”?

Answer: Thank you for your question, we decided to replace GyC by GC abbreviation for Glycerol carbonate.

Reviewer 2 Report

In this work, DESs can be used as both the solvents and orgonacatalysts for the PHUs. As a whole, this work is well done. I think it can be accepted after some minor revisions.

1. Some results can be present as figures instead of tables.

2. The Font of references should be revised.

Author Response

In this work, DESs can be used as both the solvents and orgonacatalysts for the PHUs. As a whole, this work is well done. I think it can be accepted after some minor revisions.

  1. Some results can be present as figures instead of tables.

Anwser: This comment is unclear for the authors. Generally, it is acknowledged that tables are clearer and easier to understand than figures. For this reason, we decided to keep the tables in the text. However, if the Editorial Board believes that figures (charts) would be better, we can surely make the appropriate changes.

  1. The Font of references should be revised.

Answer: Thank you we have modified the references as suggested.
